# Dynamic Normalization and Relay for Video Action Recognition

**Dongqi Cai\***
Intel Labs China
dongqi.cai@intel.com

**Anbang Yao**[†]
Intel Labs China
anbang.yao@intel.com

**Yurong Chen**
Intel Labs China
yurong.chen@intel.com

## Abstract

Convolutional Neural Networks (CNNs) have been the dominant model for video action recognition. Due to the huge memory and compute demand, popular action recognition networks need to be trained with small batch sizes, which makes learning discriminative spatial-temporal representations for videos become a challenging problem. In this paper, we present Dynamic Normalization and Relay (DNR), an improved normalization design, to augment the spatial-temporal representation learning of any deep action recognition model, adapting to small batch size training settings. We observe that state-of-the-art action recognition networks usually apply the same normalization parameters to all video data, and ignore the dependencies of the estimated normalization parameters between neighboring frames (at the same layer) and between neighboring layers (with all frames of a video clip). Inspired by this, DNR introduces two dynamic normalization relay modules to explore the potentials of cross-temporal and cross-layer feature distribution dependencies for estimating accurate layer-wise normalization parameters. These two DNR modules are instantiated as a light-weight recurrent structure conditioned on the current input features, and the normalization parameters estimated from the neighboring frames based features at the same layer or from the whole video clip based features at the preceding layers. We first plug DNR into prevailing 2D CNN backbones and test its performance on public action recognition datasets including Kinetics and Something-Something. Experimental results show that DNR brings large performance improvements to the baselines, achieving over 4.4% absolute margins in top-1 accuracy without training bells and whistles. More experiments on 3D backbones and several latest 2D spatial-temporal networks further validate its effectiveness. Code will be available at `https://github.com/caidonkey/dnr`.

## 1 Introduction

Human action recognition is a fundamental problem in video understanding, which has been studied for decades. The performance of an intelligent action recognition system depends on how well it can extract compact and discriminative features to characterize temporal evolutions of human object appearance and its motion information in videos. Early seminal methods [29, 11, 8, 46, 28, 45, 58] ubiquitously use hand-crafted features to construct spatial-temporal descriptors. In recent years, deep learning based models [70] have become the mainstream in action recognition research, mainly due to their remarkably better representation and generalization abilities compared to conventional methods, especially to model large amounts of training data.

Despite the prevalence of deep learning based methods for action recognition, learning efficient yet effective video representations is still a challenging problem. To improve the spatial-temporal

---

\* The first two authors contributed equally to the writing of the paper. [†] Corresponding author.

35th Conference on Neural Information Processing Systems (NeurIPS 2021).

representation learning of CNNs for action recognition, we investigate a new technical perspective by presenting an improved normalization design called Dynamic Normalization and Relay (DNR), learning to predict accurate layer-wise normalization parameters for any action recognition network in an input-dependent manner. We are motivated by two plain facts. On the one side, due to the problem of space-time cube data structure, the huge memory and compute demand in training a popular network for action recognition usually restricts the batch size to a much smaller range compared to the settings for image classification tasks. For instance, on Kinetics [5] the batch size is typically set to 32 video clips of length 8, while on ImageNet [44] it is set to 256 images. On the other side, existing CNN models for video action recognition use Batch Normalization (BN) [24] as a standard component to normalize feature maps learnt at each layer. Although the significance of BN has been well demonstrated in many previous action recognition works, it will easily introduce noise during the estimation of layer-wise normalization parameters when the batch size is small, degenerating the accuracy of the trained model to some extent. This problem could be alleviated by using recent normalization methods [23, 63, 26, 57, 31, 38, 67]. However, the performance gain of doing so is limited as they are primarily proposed for image recognition tasks. For example, on Kinetics with 32-frame video clips, it only brings -0.3% and 0.7% top-1 accuracy improvement to ResNet-50 C3D baseline [53] trained with a batch size of 8 and 4 video clips per GPU by replacing BN with Group Normalization (GN), as reported in [63]. We argue this is mainly because a direct extension of them from image to video domain lacks a proper mechanism to handle complicated spatial-temporal feature variations of video data. However, to the best of our knowledge, there is almost no research effort made to explore a better normalization mechanism for promoting the training of existing action recognition networks with video clip inputs.

*As an improved video normalization design for action recognition networks, our DNR considers three questions at multiple representation learning scales to alleviate the aforementioned gap:* (1) At individual frame scale, how to estimate input-dependent layer-wise normalization parameters? (2) At time scale, how to enhance the dependencies of the estimated normalization parameters for layer-specific feature representations between neighboring frames? (3) At network depth scale, how to enhance the clip-level dependencies of the estimated normalization parameters between neighboring layers? To the first question, we formulate our DNR as a dynamic normalization predication scheme [1] where normalization parameters for any layer of an action recognition network are learnt and generated dynamically both in training and inference, making them be input-dependent. To the other two questions, we bridge the DNR formulation with two interdependent normalization relay modules called cross-temporal DNR and cross-layer DNR, which are encapsulated into a light-weight recurrent structure [20]. For a certain layer, the normalization parameters learnt by cross-temporal DNR module are conditioned on the input features as well as the normalization parameters estimated from the neighboring frames based features at the same layer, while the normalization parameters learnt by cross-layer DNR module are conditioned on the input features as well as the normalization parameters estimated from the whole video clip based features at the preceding layers. On the one side, the dynamic normalization relay along the temporal axis models the layer-specific frame-level correlations of the spatial-temporal feature distributions between neighboring frames. On the other side, the dynamic normalization relay along the sequential layers considers all the hidden layers as a whole system, and estimates the video clip-level feature dynamics of the current layer by jointly considering the feature distributions of its preceding layers. Benefiting from the above complementary designs, our DNR can significantly improve the performance of existing action recognition networks via replacing their normalization layers by two types of DNR modules.

Experimental results on Kinetics and Something-Something datasets [18, 40] show that by applying DNR to prevailing 2D CNN backbones such as ResNet [19], ResNeXt [64] and BNInception [25], it brings large accuracy improvements to the baselines. We also apply DNR to 3D CNN backbones and several latest 2D CNN designs constructed with sophisticated spatial-temporal modules for further validation of its effectiveness. Thorough ablative experiments are also conducted to have a deep analysis of the proposed method.

## 2   Related Works

In this section, we summarize existing works that are related to our method, and discuss their connections and differences.

**Efficient Spatial-Temporal Modeling.** Efficient spatial-temporal modeling is a hot research topic in video action recognition. To strike a direct trade-off between network accuracy and efficiency, some works [42, 54, 37] use factorized convolutions to approximate 3D convolutions, and some works [65, 71, 15] mix 2D and 3D convolutions into a single CNN architecture. Two-path networks [16, 14, 43, 30] are another kind of efficient 3D designs. One representative is SlowFast [16] which uses a slow pathway operating at low frame rate of high image resolution to capture spatial semantics and a fast pathway operating at high frame rate of low image resolution to capture motion information. Compared to 2D CNN counterparts, the aforementioned networks mixing 2D and 3D convolutions as well as two-path networks with 3D convolutions are not runtime efficient. They are efficient only to their own motivations. In some recent works [34, 51, 47, 13, 39, 59], switchable channel grouping operations such as shift and shuffle along the temporal dimension are incorporated into either 2D or 3D CNN networks to promote the information exchange between neighboring frames. As our method improves the basic normalization strategy for training action recognition networks, it could be potentially combined with some of these efficient designs to get improved performance.

**Action Recognition with Recurrent Models.** Thanks to their natural abilities to model temporal relations within video data, Recurrent Neural Networks (RNNs) instantiated as Long Short Term Memory (LSTM) networks [20] are popularly used in action recognition. This line of research is originally explored by [3] in which an LSTM network is used to classify actions in soccer videos, taking hand-crafted features as the input. In the deep learning era, a lot of architectural extensions [69, 12, 50, 17, 48, 52, 41, 4, 55, 33] have been presented, using LSTM networks to aggregate the frame-level CNN features for video clip-level action predications. These methods follow a basic CNN-LSTM framework, using LSTM networks as the fusion module to model temporal dynamics over the frame-level features extracted by one or several pre-trained CNN models. Unlike them, we investigate a new perspective for the usage of LSTM: we bridge a light-weight LSTM structure with a dynamic normalization formulation, learning to generate frame-adaptive layer-wise normalization parameters for improving the performance of existing action recognition networks.

**Action Recognition with Attentive Models.** There are a lot of existing works which use different attention blocks to boost the spatial-temporal representation learning of CNNs for action recognition. [61] uses a self-attention block with non-local operations to capture long-range dependencies. Several recent methods [35, 32, 9] extend the attention designs of the SENet family [21, 62] to construct attention blocks for video action recognition, among which [35] addresses the attention block design for 2D CNNs while [32] and [9] are for 3D CNNs. Besides these attention blocks, dynamic convolutions [27, 68, 6] are used in [10, 36] to build their attention blocks. [36] presents a temporal adaptive module to generate video-specific temporal kernels. Compared to these action recognition methods inspired by SENet and dynamic convolutions, our method shares the similarity of modulating learnt features dynamically to improve the spatial-temporal representation learning. However, they focus on designing powerful attention blocks to replace existing blocks of an action recognition network, while our method introduces two drop-in normalization relay modules built with a light-weight recurrent structure in a dynamic video normalization perspective.

**Normalization Methods.** For modern CNNs, Batch Normalization (BN) [24] is a standard component, improving the optimization stability and accelerating the training convergence. Despite its great success, BN is prone to making an inaccurate estimation of the batched statistics when training with a small batch size, exhibiting a noticeable accuracy drop. To alleviate this issue, a lot of variant methods have been proposed. Batch Renormalization (BRN) [23] introduces two extra hyperparameters to limit the estimated batch statistics. Eval Normalization (EN) [49] presents an online strategy to estimate corrected statistics for BN. Moving Average Batch Normalization (MABN) [67] uses moving average statistics in both forward and backward propagation. These three methods concentrate on improving estimated statistics, particularly mean and standard deviation for each channel. Recurrent Batch Normalization [7] utilizes the benefits of BN to normalize the hidden states of LSTM networks. Switchable Normalization (SN) [57] learns to automatically select the best choice for each layer among a predefined set of normalizers including BN, Layer Normalization (LN) [2] and Instance Normalization (IN) [56]. Group Normalization (GN) [63] divides the channels into groups and computes the group-wise batched statistics. Adaptive Normalization (AdaNorm) [66] controls the scaling weight of LN with a simple linear transformation. Batch Kalman Normalization (BKN) [57] considers layer correlations when computing statistics of a certain BN layer. Following the basic principle of BKN to model layer correlations, DIANet [22] uses an LSTM network instead of Kalman Filtering process to improve the design of convolutional attention blocks for image classification

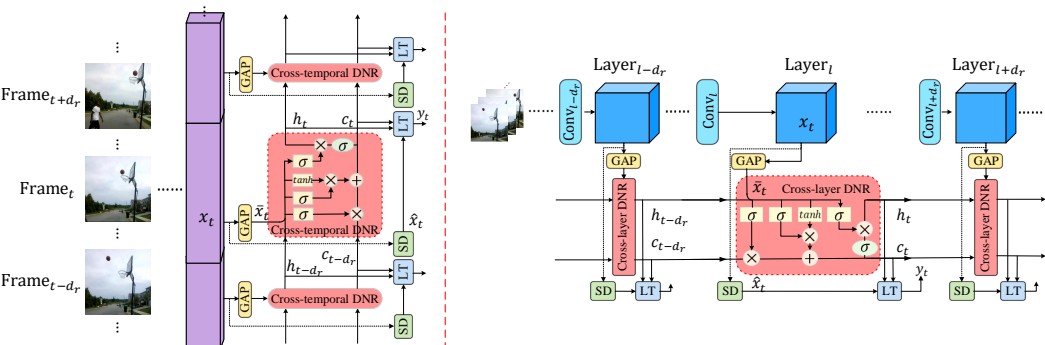

Figure 1: A schematic overview of DNR. The left figure shows cross-temporal DNR (for neighboring frames at the same layer) and the right figure shows cross-layer DNR (for the whole video clip between neighboring layers). They are encapsulated into a light-weight recurrent structure. SD/LT denotes standardization/linear transform, and the mathematical notations are clarified in the Method section. Being a drop-in design, DNR could be used to replace existing normalization layers of any action recognition network, largely improving the model training as shown in the experiments (where cross-temporal DNR and cross-temporal DNR are typically used in an interleaving manner).

tasks. However, DIANet leads to noticeable top-1 accuracy drops (over 3% even on CIFAR-100 dataset) when using its attention blocks to replace batch normalization layers. Recently, there are also some works attempting to make the normalization be sample-adaptive. Instance-Level Meta Normalization (ILM) [26] learns to predict the normalization parameters adaptively with an encoder-decoder sub-network. Attentive Normalization [31] makes an integration of feature normalization and channel-wise feature attention [21]. As discussed in the previous section, these normalization methods are mainly proposed for image recognition tasks, while this work explores the dynamic normalization mechanism to promote the training of action recognition networks at multiple scales.

## 3    Method

In this section, we first review mainstream normalization methods for video action recognition in a general formulation, and then present the insights of our method, its formulation and design details.

### 3.1    Normalization Formulation

Given a deep video model, let $x \in \mathbb{R}^{N \times C \times T \times H \times W}$ be the feature tensor extracted at a certain layer, where $N$ is the batch size, $C$ is the number of channels, $T$ is the video clip length, $H$ and $W$ are the spatial feature height and width. In the subsequent normalization, mainstream methods such as BN, IN, LN and GN perform a two-stage computation. The first stage is the standardization (SD):

$$\hat{x}_i = \frac{x_i - \mu}{\sigma}, \tag{1}$$

where $i$ is the feature index, $\mu$ and $\sigma$ denote the feature mean and standard deviation. For all methods of BN, IN, LN and GN, the input feature tensor is divided into $K$ non-overlapping sub-sets. The main difference of them lies in the feature partition which is defined along different feature dimensions. Let $\{S_1, S_2, \cdots, S_K\}$ denote the resulting feature pixel sub-sets, then their mean and standard deviation statistics could be approximated by: $\mu_k = \frac{1}{|S_k|} \sum_{x_i \in S_k} x_i$ and $\sigma_k = \sqrt{\frac{1}{|S_k|} \sum_{x_i \in S_k} (x_i - \mu_k)^2 + \epsilon}$, where $\epsilon$ is a small positive constant. After $x$ is standardized to have zero mean and unit variance, the following stage is a channel-wise linear transform (LT) to recover the feature representation ability:

$$y_i = \gamma \hat{x}_i + \beta, \tag{2}$$

where $\gamma$ and $\beta$ are learnable scale and shift parameters.

### 3.2    Dynamic Normalization and Relay

As we discussed before, existing normalization methods primarily consider image recognition tasks. Directly using them to handle video action recognition task would not likely to get satisfied

performance as video data usually incorporate complex spatial-temporal variations. Motivated by this, we present Dynamic Normalization and Relay (DNR), an improved drop-in normalization design, to boost the training of action recognition networks. Intuitively, being a normalization solution for video action recognition, it is necessary to think of feature dependencies among video frames. Accordingly, our DNR explores feature dependencies at multiple scales. On the one side, note that BN is used as the standard normalization layers in almost all high-performance action recognition networks, however it is not sample-adaptive as the learnt normalization parameters are fixed during evaluation. Recent works [21, 62, 27, 68, 6, 31, 26, 1] show that CNNs could acquire stronger representation abilities when their parameters are conditioned on the pertinent priors. Inspired by them, given a normalization operator (e.g., BN and GN, which are considered in our experiments), we first formulate our DNR as a dynamic normalization prediction scheme, learning to generate input-dependent normalization parameters of the linear transform (i.e., the second stage defined in Eq. 2) while calculating the parameters of the standardization as usual. One the other side, existing works on the fusion module designs for action recognition [69, 12, 50, 17, 48, 52, 41, 55, 33] show the effectiveness of LSTM networks in modeling motion dynamics, and recent works on the feature calibration designs for image recognition [57, 22] indicate the importance of layer correlations in learning discriminative feature representations. *Inspired by them, in our DNR, we further bridge the dynamic normalization prediction scheme with two complementary relay modules namely cross-temporal DNR and cross-layer DNR, which are the core contributions of our method.* They are encapsulated into a light-weight LSTM structure conditioned on the current input features, and the normalization parameters estimated from the neighboring frames based features at the same layer or from the whole clip based features at the preceding layers. Cross-temporal DNR enhances the dependencies of the estimated normalization parameters of layer-specific feature representations between neighboring frames, and cross-layer DNR enhances the video clip-level dependencies of the estimated normalization parameters between neighboring layers. A schematic overview of DNR is shown in Figure 1.

In the following, we detail the formulation of DNR, considering cross-temporal DNR first. For the $l^{th}$ layer, let $x_t$ denote the feature maps extracted from the $t^{th}$ frame group sampled from an input video clip and $\bar{x}_t$ be the input feature computed by applying channel-wise Global Average Pooling (GAP) operations to $x_t$. Let the scale $\gamma_t$ and the shift $\beta_t$ be the normalization parameters need to be estimated. Cross-temporal DNR is then defined as a light-weight LSTM which models the recurrent transition of normalization parameters across neighboring frames by

$$(f_t, i_t, g_t, o_t) = \phi_h(h_{t-d_r}) + \phi_x(\bar{x}_t) + b, \tag{3}$$

where $\phi(\cdot)$ is a bottleneck unit (we will detail its structure in the next sub-section) for processing the input feature $\bar{x}_t$ and the hidden state $h_{t-d_r}$ ($d_r$ is the relay distance which will be clarified later), and $b$ is the bias. $f_t, i_t, g_t, o_t$ form a set of gates to regularize the update of the LSTM by

$$c_t = \sigma(f_t) \odot c_{t-d_r} + \sigma(i_t) \odot \tanh(g_t), \tag{4}$$

$$h_t = \sigma(o_t) \odot \sigma(c_t), \tag{5}$$

where $c_t$ is the cell state, $\sigma(\cdot)$ is the sigmoid function, and $\odot$ is the Hadamard product operator. For simplicity, we set

$$\gamma_t = h_t, \quad \beta_t = c_t. \tag{6}$$

That is, the scale $\gamma_t$ and the shift $\beta_t$ are the hidden state $h_t$ and the cell state $c_t$ that need to be learnt. This simple setting makes the normalization parameters $\gamma_t$ and $\beta_t$ for the feature maps of $t^{th}$ frame group be conditioned on not only the current input feature $\bar{x}_t$ but also the previously estimated $\gamma_{t-d_r}$ and $\beta_{t-d_r}$ at the preceding $(t-d_r)^{th}$ frame group. This controlled recurrent transition across neighboring frames based feature maps provides a frame-adaptive prediction of normalization parameters and allows a feature statistics relay along the temporal axis at the same layer.

As cross-temporal DNR considers the feature distribution dependencies at the same layer between neighboring frames, cross-layer DNR further extends this formulation to consider the feature distribution dependencies of the entire video clip between neighboring layers. Specifically, in cross-layer DNR, the LSTM is unrolled along the network depth and makes the following changes in notations: (1) $\bar{x}_t$ denotes the input feature computed by applying channel-wise GAP operations to the feature maps $x_t$ extracted from the whole video clip at the $l^{th}$ layer; (2) The scale and shift parameters $\gamma_{t-d_r}$ and $\beta_{t-d_r}$ are known priors estimated at the $(l-d_r)^{th}$ layer; (3) $\gamma_t$ and $\beta_t$ are the scale and shift parameters need to be leant for the $l^{th}$ layer.

### 3.3 Implementation of DNR

Now we describe the LSTM structure used in DNR. To get improved feature learning ability and enjoy good efficiency, we modify the structure of conventional LSTM [20] leveraging the wisdom of recent attention block designs for CNN architecture research. First, following the design principle of SENet [21], we adopt a contraction-expansion bottleneck unit to process the input feature $\bar{x}_t$ and the hidden state $h_{t-d_r}$. Let $\bar{x}_t, v_t \in \mathbb{R}^{M \times C}$ be the input and the output where $C$ is the number of channels and $M$ is the number of frames (in either a frame group for cross-temporal DNR or the whole video clip for cross-layer DNR), then this bottleneck unit takes the form:

$$v_t = W_2 \delta(W_1 \bar{x}_t), \tag{7}$$

where $W_1 \in \mathbb{R}^{C \times \frac{C}{r}}$ denotes the weight matrix of a Fully Connected (FC) layer which maps the input to a low dimensional space with the reduction ratio $r$ (typically set to 4 w.r.t. the input dimension, avoiding high model complexity). $\delta(\cdot)$ denotes the Rectified Linear Unit (ReLU) activation function, and $W_2 \in \mathbb{R}^{\frac{C}{r} \times C}$ denotes the weight matrix of another FC layer which makes the output have the same number of channels as the input. Second, following the recent practice in attentive convolutional block designs [62, 22, 68], for the activation function to compute the hidden state $h_t$, we replace the original $\tanh(c_t)$ by the sigmoid function as shown in Equation 5. We empirically find this can help to learn slightly better $\gamma_t$ and $\beta_t$ with DNR.

In implementation, *we plug our two DNR modules into action recognition networks along the temporal axis and along the sequential layers interleavingly.* On the one side, the DNR module along the temporal axis models the frame-level correlations of the spatial-temporal feature distributions between neighboring frames at the same layer. Here, for the cross-temporal DNR module, each video clip is sequentially divided into a number of non-overlapping groups with the same length first, then the relay distance $d_r$ is defined as the group length (i.e., the number of frames in each group, which is computed as video clip length divided by group number). On the other side, the DNR module along the sequential layers considers all the hidden layers as a whole system, and estimates the video clip-level feature dynamics of the current layer by jointly considering the feature distributions of its preceding layers. Here, for the cross-layer DNR module, the relay distance $d_r$ is defined as the distance between two neighboring layers added with a cross-layer DNR module. Generally speaking, our DNR could be used to replace any normalization layers in popular CNN architectures for video action recognition, such as ResNet [19], ResNeXt [64] and BNInception [25] as tested in our experiments. These networks are all constructed with a block-wise design principle, and each block usually includes two or three convolutional layers. For basic blocks with skip connections, we apply the cross-temporal DNR module to the first convolutional layer and the cross-layer DNR module to the second convolutional layers between two neighboring blocks. *At the input side of each DNR pair, we additionally insert a simple channel interlacing operation [34, 51, 47, 13] along the temporal axis to strengthen local short-term feature interactions during the model training process.* It is noted that the parameters of DNR modules could be optimized simultaneously together with any action recognition network, since their computation flow is completely differentiable.

## 4 Experiments

In this section, we provide comprehensive experiments to study our method from a lot of aspects, validate its effectiveness on different video action recognition datasets, and compare its performance with other related methods.

### 4.1 Experimental Setup

**Datasets.** Four public datasets, including Kinetics-400 [5], Kinetics-200 [65], Something-Something (Sth-Sth) V1 [18] and V2 [40], are considered in the experiments. Specifically, we use Kinetics-200 to analyze the architectural design choices of two relay modules in DNR and to perform ablative studies, and use the other three datasets to testify the generalization ability of DNR. *Details of these four datasets are described in the supplemental material.*

**Implementation Details.** For fair comparisons, we choose MMaction2[1] for implementing all methods. On Kinetics-200/-400, we randomly sample an 8-frame clip with an interval of 8 from the

---
[1]https://github.com/open-mmlab/mmaction2

Table 1: Exploring different design choices for DNR. Experiments are performed on Kinetics-200 dataset using action recognition model TSN with the ResNet50 backbone. The baseline has top-1|top-5 accuracy of $72.80\%|91.59\%$. Best results (in terms of top-1 accuracy) are bolded.

(a) Inserted locations.

| Method | Location | Top-1(%) | Top-5(%) |
|---|---|---|---|
| CT | $BN_1$ | 74.52 | 92.55 |
|  | $BN_2$ | **75.52** | **92.49** |
| CL | $BN_1$ | **75.70** | **92.63** |
|  | $BN_2$ | 74.50 | 92.30 |
| CT+CL | $BN_1 + BN_2$ | 77.28 | 93.81 |
| CL+CT (DNR) | $BN_1 + BN_2$ | **78.39** | **94.08** |

(b) Relay distance.

| Method | Relay Distance ($r_d$) | Top-1(%) | Top-5(%) |
|---|---|---|---|
| CT | 1 | 74.06 | 92.11 |
|  | 2 | 74.56 | 92.33 |
|  | 4 | **75.52** | **92.49** |
|  | 8 | 75.28 | 93.13 |
| CL | 1 | **75.70** | **92.63** |
|  | 2 | 75.44 | 92.85 |

(c) LSTM structure.

| Method | Settings | | Top-1(%) | Top-5(%) |
|---|---|---|---|---|
| DNR | reduction ratio ($d = 1$) | $r = 1$ | 77.98 | 94.23 |
|  |  | $r = 2$ | 77.72 | 94.39 |
|  |  | $r = 4$ | **78.39** | **94.08** |
|  |  | $r = 8$ | 78.29 | 94.35 |
|  | depth ($r = 4$) | $d = 1$ | **78.39** | **94.08** |
|  |  | $d = 2$ | 78.00 | 94.13 |

(d) Relay mechanism.

| Method | | Relay | Top-1(%) | Top-5(%) |
|---|---|---|---|---|
| CL | | × | 72.96 | 91.55 |
|  | ✓ | FC | 72.28 | 90.83 |
|  | | SE | 75.20 | 92.93 |
|  | | LSTM | **75.70** | **92.63** |
| CT | | × | 73.40 | 91.21 |
|  | ✓ | FC | 72.78 | 91.29 |
|  | | SE | 75.04 | 92.17 |
|  | | LSTM | **75.52** | **92.49** |
| DNR | | × | 76.90 | 93.71 |
|  | ✓ | FC | 74.50 | 92.13 |
|  | | SE | 77.78 | 93.97 |
|  | | LSTM | **78.39** | **94.08** |

full-length video (unless otherwise stated) during training. The input clip is resized to $340 \times 256$ pixels and cropped to $224 \times 224$ using multi-scale cropping. Following the from-scratch training strategy in [16], we adopt cosine schedule of learning rate decaying and use a linear warm-up strategy with warm-up ratio of $0.01$ in the first 60K/128K iterations. For evaluation, we follow common practice to uniformly sample 10 clips from a video and take 3 crops of $256 \times 256$ pixels. The prediction score is averaged over all clips. On Sth-Sth V1&V2 (with shorter video durations compared to Kinetics), we follow the pre-training (on Kinetics-400) and fine-tuning strategy, and report 1 clip and center-crop testing accuracy on validation set. *Detailed implementations are in the supplemental material.*

### 4.2 Optimal Settings of DNR

We first study the optimal design choice for two core modules of DNR, i.e., cross-temporal DNR and cross-layer DNR (CT and CL for short), and the optimal relay design of the LSTM structure for DNR and its two core modules. Experiments are performed on Kinetics-200 dataset using the popular Temporal Segment Networks (TSN) [60] with the ResNet50 backbone as the test case to explore these questions in the following aspects.

**Inserted Locations.** We first consider where to place CT and CL using our default settings of LSTM structure. Table 1a summarizes the results, where $BN_1$ and $BN_2$ denote the first and second normalization layers in a bottleneck block of ResNet50, respectively. We can find that at an individual layer, CT at $BN_2$ and CL at $BN_1$ perform better compared to the other choices. When combining CT and CL, this combination also performs the best. Besides, we also tried to place CT/CL at $BN_3$ yet got marginal top-1 gain (less than $0.3$ percent). According to these experimental results, we place CL at the first BN layer and CT at the second BN layer in each block of all networks in an interleaving manner, and used it as the default settings, yielding a good accuracy and efficiency tradeoff.

**Relay Distance.** Once the inserted locations of two typed DNR modules are determined, how to set relay distance for each of them is now critical. For CT, each video clip is sequentially divided into a number of non-overlapping groups with the same length first, then the relay distance $d_r$ is defined as the number of frames in each group. Considering that the input video length is $8$, we explore relay distance from 1 to 8. From Table 1b, the best performance of CT is achieved when the relay distance is $4$. For CL, $d_r$ is defined as the distance between two neighboring layers added with a CL module.

Table 2: Main results of DNR on Kinetics-200 dataset using TSN with different backbones. Best results are bolded.

| Method | Backbone | Top-1(%) | Top-5(%) | ΔTop-1(%) |
|---|---|---|---|---|
| TSN | BNInception | 69.13 | 89.20 | |
| TSN+DNR | | **74.52** | **92.17** | **+5.39** |
| TSN | ResNet50 | 72.80 | 91.59 | |
| TSN+DNR | | **78.39** | **94.09** | **+5.59** |
| TSN | ResNet101 | 73.58 | 92.29 | |
| TSN+DNR | | **79.37** | **94.51** | **+5.79** |
| TSN | ResNeXt101 | 75.10 | 91.55 | |
| TSN+DNR | | **79.53** | **94.19** | **+4.43** |

Table 3: Comparison of DNR with different normalization methods under small training batch sizes (BS) per GPU on Kinetics-200 dataset using TSN with ResNet50. Best results are bolded.

| BS/GPU Method | 8 | 6 | 4 |
|---|---|---|---|
| BN | 72.80/91.59 | 70.93/91.03 | 67.59/89.22 |
| DNR | 78.39/94.08 | 77.66/94.11 | 75.48/93.57 |
| ΔTop-1/-5(%) | **+5.59/2.49** | **+6.73/3.08** | **+7.89/4.35** |
| GN | 67.27/88.54 | 67.35/88.42 | 67.41/88.49 |
| DNR | 77.22/93.35 | 76.24/93.15 | 75.40/92.95 |
| ΔTop-1/-5(%) | **+9.95/4.81** | **+8.89/4.73** | **+7.99/4.46** |

Table 4: Main results of DNR on different datasets. Best results are bolded.

| Dataset | Method | Backbone | Pretrain | Top-1(%) | Top-5(%) | ΔTop-1(%) |
|---|---|---|---|---|---|---|
| Kinetics-400 | TSN | ResNet50 | None | 68.17 | 87.97 | |
| | TSN+DNR | | | **73.75** | **91.44** | **+5.58** |
| | TSN | ResNet101 | | 69.50 | 88.90 | |
| | TSN+DNR | | | **74.80** | **91.99** | **+5.30** |
| Sth-Sth V1 | TSN | ResNet50 | Kinetics-400 | 17.81 | 44.62 | |
| | TSN+DNR | | | **45.58** | **75.24** | **+27.77** |
| Sth-Sth V2 | TSN | ResNet50 | Kinetics-400 | 31.35 | 62.66 | |
| | TSN+DNR | | | **58.60** | **86.21** | **+27.25** |

Since the most common number of bottleneck blocks within one stage in ResNets is 3, we explore relay distance of 1 and 2 here. From Table 1b, relaying feature distribution statistics between nearby blocks performs better than that between blocks having a larger layer distance. Therefore, we set relay distance of CT and CL to 4 and 1 respectively as the default settings.

**Architectural Choices of LSTM**. As both CT and CL are encapsulated into an LSTM structure, it is necessary to study its architectural choices. Our LSTM structure has two basic hyper-parameters: reduction ratio $r$ and the number of bottleneck units $d$. Table 1c provides the results of experiments in which we alter the settings of $r$ and $d$ separately. It can be seen that the LSTM structure reaches a compromise between accuracy and efficiency with $r = 4$ and $d = 1$ which are used as the default settings. A somewhat surprising finding is that increasing the depth of LSTM structure does not brings extra gains to model accuracy compared to the default settings. We conjecture this maybe because the optimization has already saturated with the shallow design.

**Importance of Relay Mechanism.** Since relay mechanism plays an important role in our DNR design, we conduct experiments to show the advantage of relay and compare different relay designs. The core idea of relay is to build the connections between learnt features statistics across neighboring frames and across neighboring layers. In the experiments, we first consider the case without relay design, only using dynamic normalization of our method at the current frame or layer. To some extent, this can be considered as a simplified implementation of dynamic normalization designs such as ILM [26] and AN [31]. Next, we consider relay with "FC"(fully connected) or "SE"(squeeze and excitation) [21], meaning that we use an FC or SE unit as shared attentive relay aggregating current features and previously estimated channel statistics to conduct dynamic normalization. Table 1d shows the performance comparison. We can observe: (1) Both dynamic normalization and relay are helpful to improve the top-1 accuracy of the baseline in most cases; (2) Relay with LSTM brings over 2% extra top-1 accuracy gain to dynamic normalization; (3) DNR with LSTM structure performs the best among all designs. This comparison backs up the general dynamic relay philosophy of DNR.

## 4.3 Main Results

To validate the effectiveness of our DNR method with the default settings identified above, we apply it to train TSN with a spectrum of popular 2D video action recognition backbones on different datasets. Table 2 provides the results on Kinetics-200. Clearly, DNR brings large accuracy improvements to all baseline models with BNInception [25], ResNet [19] and ResNeXt [64] backbones, under the from-scratch training setting, yielding at most 5.79% top-1 margin and at least 4.43% top-1 margin.

Table 5: Performance comparison of applying DNR to 3D action recognition networks with different backbones on Kinetics-200 dataset. Best results are bolded.

| Method | Backbone | Top-1(%) | Top-5(%) | ΔTop-1(%) |
|---|---|---|---|---|
| R(2+1)D
R(2+1)D+DNR | ResNet34 | 71.77
**74.22** | 91.19
**92.25** | **+2.45** |
| SlowFast 8×8
SlowFast 8×8+DNR | ResNet18 | 77.37
**79.13** | 93.48
**93.89** | **+1.76** |

Results on Kinetics-400 and Sth-Sth V1&V2 are given in Table 4. We can see the improvements of DNR on Kinetics-400 and Kinetics-200 are very similar, showing its good generalization ability to a larger dataset. Significant improvements are also achieved on Sth-Sth V1&V2, demonstrating the transfer learning ability of DNR to enhance the performance of action recognition networks.

## 4.4 Ablative Studies

We further provide a number of ablative studies to have a deep analysis of our DNR method.

**Impacts of Normalization Methods and Batch Size Settings.** Being a drop-in design, DNR could be applied to other popular normalization methods besides BN. In image recognition field, GN [63] is known to have much better capability to combat small batch size training problem compared to BN. Accordingly, we also conduct experiments on Kinetics-200 to explore the effectiveness of applying DNR to GN besides BN, as well as its advantage in small batch size training. For a fair comparison, we adopt the linear learning rate scaling rule commonly used in GN to adapt to batch size changes. Specifically, we make the learning rate be proportional to the batch size ($0.1 \times N/8$, where N is the batch size per GPU). For instance, under the batch size of $8/6/4$, the learning rate is initialized to $0.1/0.075/0.05$ and decayed with a cosine scheduling. We also use a linear warm-up strategy with warm-up ratio of $0.01$ in the first 60K iterations. Table 3 compares the performance of applying DNR to BN and GN with different batch sizes. As can be seen, DNR outperforms the baseline method with consistently large margins (at least $5.59\%$ gain and at most $9.95\%$ gain to top-1 accuracy). Moreover, to BN, DNR achieves larger margins when reducing mini-batch size per GPU from $8$ to $4$ video clips, showing its advantage in resource constrained training scenarios. Compared to the results on BN with DNR, a reverse performance improvement trend can be observed on GN with DNR. This is because GN performs better in small batch size training scenarios while performing worse in large batch size training scenarios. *We also compare DNR with dynamic normalization (without relay) in Table 1d.*

**Applying DNR to 3D CNNs.** Although 3D CNNs generally have better spatial-temporal feature modeling capabilities than 2D CNNs, we also apply DNR to 3D CNN backbones to further evaluate its potential in enhancing spatial-temporal representation learning. Restricted by our available computational resource, we choose R(2+1)D [54] with ResNet34 and SlowFast [16] with ResNet18 as the baseline models, which are relatively light-weight in 3D CNNs. As shown in Table 5, DNR brings around $2\%$ top-1 accuracy boosts for two baseline models, demonstrating its efficacy to promote the performance of 3D CNNs besides 2D ones.

**Applying DNR to State-of-the-art Methods.** In order to validate the capability of DNR to boost action recognition networks with advanced temporal modules, we particularly perform another set of experiments on Kinetics-400 dataset. In the experiments, besides TSN [60], we also choose TSM [34], TANet [36] and TDN [59] as the baseline models, which are three latest 2D attentive designs for spatial-temporal modeling. We apply DNR to these four baselines for performance evaluation under the exactly same training and test settings. Table 6 shows the results, and also provides a comparison of DNR with some state-of-the-art methods including 3D and 2D CNNs. Comparatively, TSN+DNR is $2.7\%$ better than TSN (with $71.65\%$ top-1 accuracy), TSM+DNR is $1.1\%$ better than TSM (with $73.78\%$ top-1 accuracy), TANet+DNR is $0.65\%$ better than TANet (with $76.28\%$ top-1 accuracy), and TDN+DNR is $0.71\%$ better than TDN (with $76.34\%$ top-1 accuracy). Note these four baseline models contain three types of advanced action recognition models with 2D CNNs. Specifically, TSN [60] is the very basic temporal module, TSM [34] uses an improved temporal module with shift operations across neighboring frames, while more advanced attentive temporal modules are used in TANet [36] (built with dynamic convolutions) and TDN [59] (encoding frame differences into the attentive design). Particularly, TANet and TDN are two of the latest top-performing 2D attentive

Table 6: Comparison of our method with state-of-the-art methods on Kinetics-400 dataset. "∘" and "∗" refer to our reproduced models using MMaction2 and public code released by the authors, respectively. Best results are bolded.

| Model | Backbone | Pretrain | Frames | GFLOPs×Views | Top-1(%) | Top-5(%) |
|---|---|---|---|---|---|---|
| SlowFast [16] | ResNet50 | None | (4+32)×10×3 | 36.1×30 | 75.60 | 92.1 |
| SlowFast [16] | ResNet101 | None | (8+32)×10×3 | 106×30 | 77.90 | 93.20 |
| NL SlowFast [16] | ResNet101 | None | (16+64)×10×3 | 234×30 | 79.80 | 93.90 |
| X3D [15] | X3D-XL | None | 16×10×3 | 48.4×30 | 79.10 | 93.90 |
| SmallBigNet [30] | ResNet50 | None | 8×10×3 | 57×30 | 76.30 | 92.50 |
| TSN [60] | ResNet50 | ImageNet | 25×1×10 | 4.11×250 | 70.60 | 89.26 |
| TSM [34] | ResNet50 | ImageNet | 8×10×3 | 33×30 | 74.10 | N/A |
| TEINet [35] | ResNet50 | ImageNet | 8×10×3 | 33×30 | 74.90 | 91.80 |
| TEA [32] | ResNet50 | ImageNet | 16×10×3 | 70×30 | 76.10 | 92.50 |
| TAM [14] | bLResNet50 | Kinetics | 48×3×3 | 93.4×9 | 73.50 | 91.20 |
| TANet [36] | ResNet50 | ImageNet | 8×10×3 | 36×30 | 76.09 | 92.30 |
| TDN [59] | ResNet50 | ImageNet | 8×10×3 | 36×30 | 76.60 | 92.80 |
| TSN∘ | ResNet50 | None | 8×10×3 | 33×30 | 71.65 | 90.17 |
| TSN+DNR | ResNet50 | None | 8×10×3 | 33×30 | **74.35** | **91.68** |
| TSM∘ | ResNet50 | ImageNet | 8×10×3 | 33×30 | 73.78 | 91.32 |
| TSM+DNR | ResNet50 | ImageNet | 8×10×3 | 33×30 | **74.88** | **91.99** |
| TANet∘ | ResNet50 | ImageNet | 8×10×3 | 36×30 | 76.28 | 92.46 |
| TANet+DNR | ResNet50 | ImageNet | 8×10×3 | 36×30 | **76.93** | **92.81** |
| TDN∗ | ResNet50 | ImageNet | 8×10×3 | 36×30 | 76.34 | 92.63 |
| TDN+DNR | ResNet50 | ImageNet | 8×10×3 | 36×30 | **77.05** | **92.98** |

temporal modules, in this point of view, the performance improvement of our DNR is decent as DNR is merely added to normalization layers. Generally, DNR performs the best among 2D CNNs under the same settings.

**More Ablative Experiments.** *In the supplemental material, we provide more ablative experiments, showing:* (1) Bidirectional LSTM design can slightly improve cross-temporal DNR; (2) The number of extra FLOPs introduced by DNR is almost negligible, and the extra runtime latency is not negligible but is acceptable; (3) DNR learns better spatial-temporal features by visualization comparisons.

## 4.5   Limitations of DNR

Despite that DNR shows promising performance improvements to a lot of action recognition networks, its limitations are in two aspects. Firstly, in the experiments, we provide the comparison of computational cost (in terms of both FLOPs and runtime inference speed) of different baseline models without and with DNR, showing although DNR introduces almost negligible extra FLOPs to different baselines, it introduces extra latency at inference stage. Secondly, DNR has three critical hyperparameters, namely the layer location, the relay distance and the LSTM structure for setting cross-temporal and cross-layer DNR modules. Although we provide relatively thorough experiments on Kinetics-200 dataset with the TSN+ResNet50 baseline to study the setting for each of them, and apply the resulting settings to all baseline models tested in our experiments, it is not the optimal settings to different action recognition networks. Besides, the potential of applying DNR to emerging new types of deep action recognition networks is reserved as a future extension.

## 5   Conclusions

In this paper, we present DNR, a drop-in normalization method, to promote the spatial-temporal representation learning of CNNs for video action recognition. The core contributions of DNR are cross-temporal and cross-layer dynamic normalization and relay designs, learning to generate frame-adaptive layer-wise normalization parameters at both training and runtime. The efficacy of DNR was validated by thorough experiments on several public action recognition datasets.

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
