# Supplementary Material for "Dynamic Normalization and Relay for Video Action Recognition"

**Dongqi Cai***
Intel Labs China
dongqi.cai@intel.com

**Anbang Yao**[†]
Intel Labs China
anbang.yao@intel.com

**Yurong Chen**
Intel Labs China
yurong.chen@intel.com

## A  Datasets

We evaluate our DNR method on four action recognition datasets following standard training/testing splits and protocols.

**Kinetics-400** [1] is a large scale action recognition dataset with trimmed video clips of around 10-second durations. It is collected from realistic YouTube videos, which covers 400 categories of human activities. In total, it contains around 240K training videos and 20K validation videos. Because of the expirations of some YouTube video links, the sizes of kinetics-400 dataset copies used in existing works may be different. The specific version of Kinetics-400 used in our experiments contains $240, 171$ training videos and $19, 772$ validation videos, which is almost the same as that used in official MMaction2 [1] (with $240, 436$ training and $19, 796$ validation videos).

**Kinetics-200** [13] is a public subset of Kinetics-400, consisting of top 200 human activity categories in Kinetics-400 according to the number of video samples per human activity category. For each category, there are $400$ training videos and $25$ validation videos which are randomly sampled from training and validation set of Kinetics-400 respectively, resulting in 80K training videos and 5K validation videos in total. Considering the efficiency and generalization tradeoff, we first use Kinetics-200 to conduct comprehensive ablative studies, and then use Kinetics-400 for performance comparison of our method with some state-of-the-art methods.

**Something-Something V1** and **V2** (Sth-Sth V1&V2) [5, 8] are two datasets focusing on fine-grained actions interacted with daily objects, covering 174 action categories. Videos in Sth-Sth V1&V2 have a duration of around 50 frames. Sth-Sth V1 comprises around 86K training and 12K validation videos. As for the resolution, it has 100-pixel height and variable widths. Sth-Sth V2 consists of around 169K training and 25K validation video clips, and the resolution is increased to have a height of 240 pixels.

## B  Implementation Details

All the experiments on Kinetics-200 use from-scratch training as in [4]. Specifically when training Kinetics-200/-400 from scratch, we adopt the cosine schedule of learning rate decaying with an initial learning rate of $0.1$. We also use a linear warm-up strategy with warm-up ratio of $0.01$ in the first 60K/128K iterations for Kinetics-200/-400, respectively. The total training epochs for Kinetics-200/-400 are 120/196, respectively. We use the momentum of $0.9$, the weight decay of $1e-4$ and the dropout of $0.4$. For evaluation, we follow common practice to uniformly sample 10 clips from a video and take 3 crops of $256 \times 256$ pixels for each clip. The prediction score is averaged over all clips. When applying our DNR to three latest 2D action recognition networks built with advanced temporal modules [6, 7, 11], we use the same training configurations as those are publicly available

---

\* The first two authors contributed equally to the writing of the paper. [†] Corresponding author.
[1] https://github.com/open-mmlab/mmaction2

35th Conference on Neural Information Processing Systems (NeurIPS 2021).

Table 1: Impacts of different cross-temporal relay directions using TSN with the ResNet50 backbone for experiments on Kinetics-200 dataset. The baseline has top-1|top-5 accuracy of 72.80%|91.59%.

| Method | Relay Distance | Relay Directions | Top-1(%) | Top-5(%) |
|--------|---------------|------------------|----------|----------|
| CT | $d_r$=2 | Uni-direction | 74.56 | 92.33 |
| | | Bi-direction | **75.56** | **92.93** |
| | $d_r$=4 | Uni-direction | **75.52** | **92.49** |
| | | Bi-direction | 74.92 | 92.61 |

in MMaction2. *We provide example codes to reproduce experimental results on Kinetics datasets when applying DNR to the TSN baseline [12] with 2D CNN backbones.*

For Sth-Sth V1&V2 (with shorter video durations compared to Kinetics), we initialize models with Kinetics-400 pre-trained weights and fine-tune models for 50 epochs. The initial learning rate is 0.005 and decays by 0.1 at epoch 20 and 40. We set the momentum to 0.9, the weight decay to $5e-4$ and the dropout to 0.5. Following common practice on Sth-Sth datasets, we report 1 clip and center crop testing accuracy on validation set.

Recall that at the input side of each DNR pair, we additionally insert a simple channel interlacing operation [6, 10, 9, 3] along the temporal axis to strengthen local short-term feature interactions during the model training process. Specifically, channel interlacing is implemented by expanding the feature maps along the temporal dimension first, then dividing all channels into 8 groups, and finally shifting the first/second channel group to left/right by one step along the temporal dimension.

## C   Relay Direction

We have shown the ablative study on relay distances in Table 1b of the paper. For cross-temporal DNR (CT), here we explore the effect of relay direction. Since feature dynamics of current frame may have direct correlations with previous and subsequent frames, we try to relay the dynamic dependencies along temporal axis in both directions. Specifically, we average the relay dynamics from both directions. We can see from Table 1 that bi-directional setting performs better when decreasing the relay distance $d_r$ (i.e., the number of frames in each group, as defined in the Method section of the paper) from 4 to 2. However, the performance gain can be ignored when comparing with uni-directional relay at $d_r = 4$. Considering efficiency, we use $d_r = 4$ and uni-directional relay as the default settings for cross-temporal DNR in our experiments.

## D   Efficiency of DNR

**Computational Complexity.** Regarding the computational complexity of DNR, the extra parameters and FLOPs are mainly from the LSTM structure. Denote the number of channels in the inserted block as $C$, an LSTM structure with reduction ratio $r = 4$ and depth $d = 1$ has $5/2 \times C^2$ parameters. Taking our default settings of applying DNR to all bottlenecks in ResNet50 backbone as an example, DNR brings 4.1M extra parameters and 0.006G extra FLOPs to the baseline model, while yielding over 5% top-1 accuracy boost on Kinetics-400 dataset. A detailed comparison of FLOPs for our method and some state-of-the-art methods is given in Table 6 of the paper. By using DNR to replace existing normalization layers, the number of extra FLOPs introduced by DNR is almost negligible compared to different baseline models.

**Runtime Inference Speed.** As the runtime inference speed is critical for real deployment of action recognition models, we also provide a comparison of runtime inference speed of different baseline models without and with our DNR. For a fair comparison, we adopt benchmark script of MMaction2 under the exactly same settings for each pair of trained models reported in Table 6 of the paper. We use a server with an NVIDIA GeForce RTX 2080Ti GPU and an Intel Xeon Gold 6240 CPU Processor, and test each model with 2000 video clips from Kinetics-400 dataset, and report the average speed (videos per second) in Table 2 (basic experimental settings are the same to those used for Table 6 of the paper). It can be seen that the runtime inference speed of our models is generally slower than that of the baselines to some degree. However, for each pair of models trained without

Table 2: Runtime inference speed comparison of different baseline models trained without and with our DNR.

| Model | Backbone | Pretrain | Frames | GFLOPs×Views | Top-1(%) | Top-5(%) | Inference Speed(videos/second) |
|---|---|---|---|---|---|---|---|
| TSN | ResNet50 | None | 8×10×3 | 33×30 | 71.65 | 90.17 | 10.9 |
| TSN+DNR | ResNet50 | None | 8×10×3 | 33×30 | 74.35 | 91.68 | 7.6 |
| TSM | ResNet50 | ImageNet | 8×10×3 | 33×30 | 73.78 | 91.32 | 9.5 |
| TSM+DNR | ResNet50 | ImageNet | 8×10×3 | 33×30 | 74.88 | 91.99 | 7.0 |
| TANet | ResNet50 | ImageNet | 8×10×3 | 36×30 | 76.28 | 92.46 | 2.8 |
| TANet+DNR | ResNet50 | ImageNet | 8×10×3 | 36×30 | 76.93 | 92.81 | 1.8 |
| TDN | ResNet50 | ImageNet | 8×10×3 | 36×30 | 76.34 | 92.63 | 2.1 |
| TDN+DNR | ResNet50 | ImageNet | 8×10×3 | 36×30 | 77.05 | 92.98 | 1.7 |

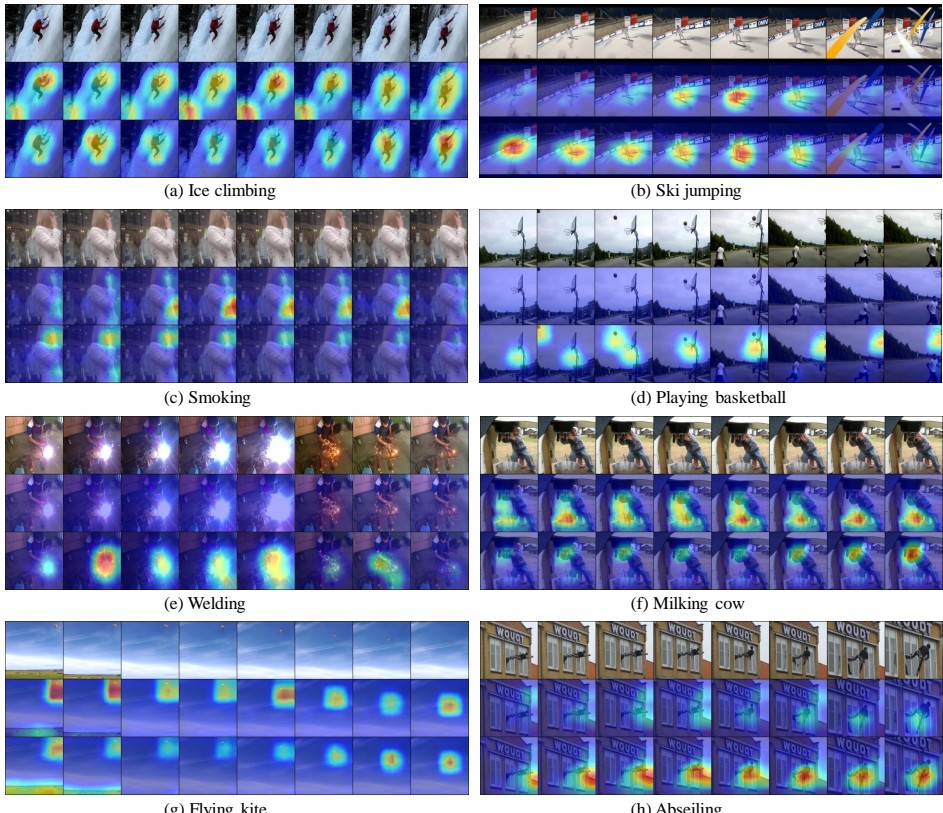

(a) Ice climbing      (b) Ski jumping

(c) Smoking      (d) Playing basketball

(e) Welding      (f) Milking cow

(g) Flying kite      (h) Abseiling

Figure 1: Visualization comparisons of activation maps with Grad-Cam++ [2]. From up to bottom: original input sequence; baseline; DNR. DNR tends to localize consistent and accurate motion related regions in different video examples, especially when the scenario is with high color contrast or subtle motions. Best viewed in color.

and with our DNR, the runtime inference speed is still at a similar level. Considering promising accuracy gains, the extra latency of our DNR to the baselines is acceptable.

## E  Visualization

To intuitively analyze the efficacy of DNR, we use Grad-CAM++ [2] to visualize the class activation maps of ResNet50 models trained on Kinetics-200. Figure 1 shows several visualization examples covering different actions. As shown, the DNR model can generally learn consistent and accurate motion-related regions. Specifically, in "Playing basketball" sequence as shown in Figure 1(d), the DNR model pays attention to the basket and ball centered regions although they have varying distances over time and the target areas are small. The visualization comparisons indicate that the

baseline model with only spatial convolutions fails to activate the motion-salient regions, while the model trained with DNR is able to consistently and accurately localize the spatial-temporal action-relevant regions, thanks to DNR's ability to enhance spatial-temporal video feature learning.