# OpenReview forum: "Dynamic Normalization and Relay for Video Action Recognition"
_NeurIPS.cc/2021/Conference — NeurIPS 2021 Poster_

### Official Review · Reviewer_kHhy · 2021-07-07

**Rating:** 7
**Confidence:** 4

**Summary:**

This paper proposes a novel dynamic batch normalization and relay (DNR) approach regarding the sequential information in videos. The authors have demonstrated extensive experiments and shown the effectiveness of the proposed DNR for a number of advanced backbones for video action recognition.

Apart from some technical confusions and questions, it is an interesting paper and the idea of dynamic batch normalization may be useful to the community.

**Limitations And Societal Impact:**

Have not addressed adequate limitations and potential negative societal impact.

It can be done such as by providing the inference speed of with and w/o DNR to demonstrate how much latency is introduced to the inference stage. Also, there are a number of hyperparameters in DNR which may cause differences on different backbones.

**Main Review:**

This paper proposes a novel dynamic batch normalization and relay (DNR) approach regarding the sequential information in videos. The authors argue that traditional batch normalization is not frame-adaptive and ignores dependencies of the estimated normalization parameters between neighboring frames. Then the proposed DNR contains cross temporal DNR (CT) and cross layer DNR (CL) to handle the above problems.

The paper is well presented and the idea of DNR for video action recognition is novel. The authors have demonstrated extensive experiments and shown the effectiveness of the proposed DNR for a number of advanced backbones for video action recognition. Apart from good results in the paper, I have few technical questions on the paper below:

1. P2 L54 mentions that the normalization parameters are generated dynamically both on training and inference. The authors also use the word “meta learning” in the abstract. Traditional meta learning requires extra training or fine-tuning procedure during the inference. However, I did not see that specific description on extra training on inference for specifying normalization parameters. I suppose inference stage directly uses the trained parameters and does not contain extra training during inference. If this is exactly what you are doing, I think the word “meta learning” and meta LSTM is somewhat misleading.

2. P7 L256 “relay distance is the number of sampled frames”. What does this mean? Does this correspond to the num_goups in _TBatchHyperNorm class in the provided codes?

3. P8 L305 “We compare DNR with dynamic normalization (without relay) in Table 1d.” Does no relay mean that (e.g., for CT) operate batch normalization simply on different frames independently?

4. Why not also perform three crops evaluation on SthV1 & SthV2 ? I understand 1 centre-crop is more efficient but three crops evaluation is able to give more convincing results to the audience.

5. It is good to see DNR introduces very limited extra FLOPs to the model in Table 6. How about the inference speed compared to no DNR? In some real applications, inference speed is more important than the introduced FLOPs.

6. In table 3, the authors experiment with different batch size performances. Is the [linear scaling rule](https://research.fb.com/wp-content/uploads/2017/06/imagenet1kin1h5.pdf) used in this experiment since the different batch size requires different learning rates? The authors should clarify the learning rate used for each batch size experiment.

**Time Spent Reviewing:**

5

---

> ### Author Response · Authors · 2021-08-10
> **Responses to Official Review by Reviewer kHhy**
>
> Thank you so much for the constructive comments and the recognition of the novelty, the presentation and the experiments of our work. Please see our below responses to your questions in **“Main Review”** and comments in **“Limitations And Societal Impact”** one by one.
>
> 1.**To your question** “P2 L54 mentions...I suppose inference stage directly uses...If this is exactly what you are doing, I think the word “meta learning” and meta LSTM is somewhat misleading.”
>
> **Our responses**: **(1)** For our method, at inference stage, the linear transform parameters of normalization layers are dynamically generated from the trained LSTM units, taking the feature maps of different video frames and the estimated linear transform parameters between neighboring frames (to cross-temporal DNR, i.e., CT) or between neighboring layers (to cross-layer DNR, i.e., CL) as the inputs. **(2)** Indeed, traditional meta learning requires extra training or fine-tuning procedure during the inference. For our DNR, we used the word “meta”|”meta learning” mainly referring to the dynamic strategy which utilizes a shared lightweight LSTM unit to generate frame-adaptive normalization parameters conditioned on the input priors (i.e., the current feature maps and the previous normalization parameters as discussed above). To a certain degree this is akin to Meta Networks [1] and HyperNetworks [2] which use a network to generate task-adaptive weights for another network. We sincerely thank you for pointing out this potentially misleading issue of the word “meta”|”meta learning”, and we will drop it and replace it by “frame-adaptive|attentive learning” during paper revision.
>
> 2.**To your question** “P7 L256 “relay distance is the number of sampled frames”. What does this mean? Does this correspond to the num_groups in _TBatchHyperNorm class in the provided codes?”
>
> **Our responses**: Here, for cross-temporal DNR (i.e., CT in Table 1b.), each video clip is sequentially divided into a number of non-overlapped groups with the same length first, then relay distance is defined as the group length (i.e., the number of frames in each group, which is computed as video-clip-length/group-number). We are sorry for such a confusion and will make it more clear during paper revision.
>
> 3.**To your question** “P8 L305 “We compare DNR with dynamic normalization (without relay) in Table 1d.” Does no relay mean that (e.g., for CT) operate batch normalization simply on different frames independently?”
>
> **Our responses**: “without relay” means using a linear layer to dynamically generate parameters $\gamma$ and $\beta$ of batch normalization, taking the current feature maps of different frames individually as the inputs. It performs better than vanilla batch normalization (shown in the caption of Table 1) in top-1 accuracy as can be seen from Table 1d.
>
> 4.**To your question** “Why not also perform three crops evaluation on SthV1 & SthV2 ? I...but three crops evaluation is able to give more convincing results to the audience.”
>
> **Our responses**: **(1)** We adopted center-crop evaluation for performance comparisons on SthV1 & SthV2, following recent state-of-the-art methods such as TEINet [8], TANet [9] and TIN [11] (see Table 2 in the supplementary materials) which use this evaluation strategy, aiming to have a fair comparison with them. **(2)** By using three crops evaluation, slightly better results are obtained by both baselines and our method, and our method shows similar accuracy improvements to baselines compared to center-crop evaluation, as can be seen from the results shown in below two Tables.
>
>
> Dataset | Method | Backbone | Pretrain | Top-1(\%)  | Top-5(\%)  | $\Delta$Top-1(center-crop, \%)  | Top-1(\%)  | Top-5(\%)  | $\Delta$Top-1(3-crop, \%)
>  :--: | :--: | :--: | :--: | :--: | :--: | :--: | :--: | :--: | :--:
>  Sth-Sth V1 | TSN | ResNet50 | ImageNet | 17.46  | 44.85  |  - | **18.31**  | **45.82** | -
>  Sth-Sth V1 | Ours| ResNet50 | ImageNet | 48.22  | 78.55  | +30.76 | **49.35**  | **79.23** | **+31.04**
>  Sth-Sth V2 | TSN | ResNet50 | ImageNet | 31.35  | 62.66  |  - | **31.79**  | **63.21** | -
>  Sth-Sth V1 | Ours| ResNet50 | ImageNet | 61.62  | 87.59  | +30.27 | **62.64**  | **88.56** | **+30.85**
>
>
> Model | Backbone | Pretrain | Frames$\times$Views | Sth-Sth V1 Top-1(\%) | Sth-Sth V2 Top-1(\%)  | Frames$\times$Views | Sth-Sth V1 Top-1(\%) | Sth-Sth V2 Top-1(\%)
> :--: | :--: | :--: | :--: | :--: | :--: | :--: | :--: | :--:
> TSM | ResNet50 | Kinetics | 8$\times$1 | 45.58 | 59.13 | **8$\times$3** | **47.18** | **60.43**
> TSM+**DNR** | ResNet50 | Kinetics | 8$\times$1 | 48.52 | 61.18 | **8$\times$3** | **50.08** | **62.58**
> TANet | ResNet50 | Kinetics | 8$\times$1 | 46.28 | 60.35 | **8$\times$3** | **47.88** | **61.48**
> TANet+**DNR** | ResNet50 | Kinetics | 8$\times$1 | 49.16 | 61.92 | **8$\times$3** | **50.72** | **63.01**
>
>
> 5.**To your question** “It is good to see...in Table 6. How about the inference speed compared to no DNR? In some real applications, inference speed is more important than the introduced FLOPs.”
>
> **Our responses**: **(1)** We agree that the runtime inference speed is more important the introduced FLOPs for real deployment of action recognition models. **(2)** You are referred to **our second set of responses to Reviewer zNtD** for the reason why we originally reported FLOPs instead of runtime inference speed to compare computational cost of different methods on Kinetics-400, and an extra comparison of runtime inference speed of different baseline models w/o and w/ our DNR.
>
> 6.**To your question** “In table 3, ...Is the linear scaling rule used in this experiment since the different batch size requires different learning rates? The authors should clarify the learning rate used for each batch size experiment.”
>
> **Our responses**: You are correct. For a fair comparison, we adopted the linear learning rate scaling rule commonly used in GN and [3-4] to adapt to batch size changes. Specifically, we make the learning rate be proportional to the batch size (0.1$\times N/8$, where N is the batch size per GPU), e.g., for the batch size of 8/6/4, the learning rate is initialized to 0.1/0.075/0.05 and decayed with a cosine scheduling. We also use a linear warm-up strategy with warm-up ratio of 0.01 in the first 60K iterations.
>
> 7.**To your comments** regarding the inadequate limitations and potential negative societal impact of our work.
>
> **Our responses**: We follow your constructive suggestions and analyze the limitations of our work in two aspects. **(1)** We provide the comparison of computational cost (in terms of both FLOPs and runtime inference speed of different baseline models w/o and w/ our DNR, showing although DNR introduces almost negligible extra FLOPs to different baselines, it introduces 9.5%-16.5% extra latency at inference stage (for runtime speed in terms of video per second, and for runtime speed in terms of second per video, the extra latency is 10.5%-19.6%). **(2)** Our DNR has three critical hyperparameters, namely the layer locations, the relay distance and the LSTM structure for setting CT and CL designs. Although we provide relatively thorough ablative studies with the TSN+ResNet50 baseline to analyze the effect of each of them, and apply the resulting setting to all baseline models, it is not the optimal setting to different convnet backbones. Besides, the potential of applying DNR to emerging backbones such as ViT and MLP designs is reserved as a future extension.
>
> **Finally**, regarding more improvements to the explanations, the discussions and the experiments that we have made, you are referred to our top-level comments titled **“The Summary of Our Responses”** and our responses to the other reviewers.
>
> [1] D. Ha, A. M. Dai, and Q. V. Le. Hyperneteworks. In ICLR, 2017.
>
> [2] T. Munkhdalai and H. Yu. Meta networks. In ICML, 2017.
>
> [3] L. Bottou, F. E. Curtis, and J. Nocedal. Optimization methods for large-scale machine learning. arXiv:1606.04838, 2016.
>
> [4] P. Goyal, P. Dollar, R. Girshick, P. Noordhuis, L. Wesolowski, A. Kyrola, A. Tulloch, Y. Jia, and K. He.
> Accurate, large minibatch SGD: Training ImageNet in 1 hour.

---

> > ### Comment · Reviewer_kHhy · 2021-08-18
> > **Feedback to authors' response**
> >
> > Thanks for the thorough response provided by the authors. My initial main concern of the paper is the real-time latency added by the module. The authors have already stated there is  "it introduces 9.5%~16.5% extra latency at inference stage" in the rebuttal, which is acceptable. Since authors have validated the generalization on various backbones and several current benchmarking datasets, the efficacy of proposed module can be validated. I hope this information such as extra latency and FLOPs can be added to the revised version either in the main text or the supplementary materials, which is very important for researchers to follow.
> >
> > According to some questions I asked in the initial review stage, the authors explain them carefully. I think this approach is novel and different from most of current approaches. This may be a interest to the community. I suggest to accept this paper and raise my score.

---

### Official Review · Reviewer_zNtD · 2021-07-15

**Rating:** 6
**Confidence:** 4

**Summary:**

This paper presents a Dynamic Normalization and Relay (DNR) module to augment the spatial-temporal representation learning for action recognition. The proposed DNR module can ease the effect of small batch size for video action recognition therefore improve the performance.

**Limitations And Societal Impact:**

Yes.

**Main Review:**

Strengths:
- The authors proposed a DNR module to dynamic normalize each layer according to the input sample, which can ease the weakness of small batch size when training a video action recognition networks.
- Extensive  ablation studies are conducted to show the effectiveness of their method.

Weaknesses:
- How the standardization parameters \mu and \sigma  are calculated is not very clear in the paper, is that the same as BN?
- The runtime of their method compared with other methods should be given apart from FLOPs, because they use LSTM structures in their module, FLOPs cannot represent the true performance of the model

**Time Spent Reviewing:**

4

---

> ### Author Response · Authors · 2021-08-10
> **Responses to Official Review by Reviewer zNtD**
>
> Thank you so much for the constructive comments and the recognition of the novelty and the experiments of our work. Please see our below responses to your questions in “Weaknesses” one by one.
>
> 1.**To your question** “How the standardization parameters $\mu$ and $\sigma$ are calculated is not very clear in the paper, is that the same as BN?”
>
> **Our responses**: As we stated in line#158 to line#159, our DNR method is proposed for learning to generate frame-adaptive normalization parameters of the linear transform (i.e., the second stage of a given normalization operator, defined in Equation 2). As for the parameters $\mu$ and $\sigma$ of the standardization (i.e., the first stage of a normalization operator, defined in Equation 1), they are calculated according to their corresponding definitions (shown in line#143) of a given normalization operator such as BN and GN which are both considered in our experiments (see Table 3). We will make it more clear during paper revision.
>
> 2.**To your request** “The runtime of their method compared with other methods should be given apart from FLOPs, because they use LSTM structures in their module, FLOPs cannot represent the true performance of the model.”
>
> **Our responses**: **(1)** We agree that the runtime inference speed is critical for real deployment of action recognition models. In the comparison of our method with state-of-the-art methods on Kinetics-400 (Table 6 of the main paper), we followed a common evaluation protocol and reported FLOPs instead of runtime inference speed to compare computational cost of different methods as most state-of-the-art methods did not report runtime inference speed. **(2)** Following the constructive suggestion/request by you and the other two reviewers, we provide a comparison of runtime inference speed of different baseline models w/o and w/ our DNR. For a fair comparison, we adopt benchmark script of MMaction2 under the exactly same settings for each pair of models. We use a server with an NVIDIA GeForce RTX 2080Ti GPU and an Intel Xeon Gold 6240 CPU Processor, and test each model with 2000 video clips from Kinetics-400 dataset, and report the average speed (second per video) in below Table (basic experimental settings are the same to those used for Table 6). It can be seen that the runtime inference speed of our method is about 0.835~0.905$\times$ to that of the baselines. Though the extra cost in terms of runtime inference speed of our method to the baselines is higher than that in terms of FLOPs, it is still acceptable for the accuracy gain.
>
> |  Model |  Backbone | Pretrain |   Frames      | GFLOPs$\times$Views | Top-1(%) | Top-5(%) | Inference Speed (sec/video) |
>   | :--: | :-------: | :----------: |   :-----------:     | :------------------------: | :------------: | :------------: | :----------------------------: |
>  | TSN$^{\circ}$    | ResNet50  | None    |8$\times$10$\times$3 |      33$\times$30           |    71.65    |    90.17   |      0.092 |
>  | TSN+**DNR** | ResNet50 | None | 8$\times$10$\times$3 | 33$\times$30 | **74.35** | **91.68** | **0.110** |
>  | TSM$^{\circ}$  |  ResNet50  |  ImageNet  | 8$\times$10$\times$3 | 33$\times$30 | 73.78 | 91.32 | 0.105 |
>  | TSM+**DNR** | ResNet50 | ImageNet | 8$\times$10$\times$3 | 33$\times$30 | **74.88** | **91.99** | **0.119** |
>  | TANet$^{\circ}$ | ResNet50 | ImageNet | 8$\times$10$\times$3  | 36$\times$30 | 76.28 | 92.46 | 0.357 |
>  | TANet+**DNR** | ResNet50 |ImageNet|8 X 10 X 3|36$\times$30 | **76.93** | **92.81** | **0.400** |
>  | TDN$^\ast$ | ResNet50 | ImageNet | 8$\times$10$\times$3 | 36$\times$30 | 76.34 | 92.63 | 0.476 |
>  | TDN+**DNR** | ResNet50 | ImageNet | 8$\times$10$\times$3 | 36$\times$30 | **77.05** | **92.98** | **0.526** |
>
>
> **Finally**, regarding more improvements to the explanations, the discussions and the experiments that we have made, you are referred to our top-level comments titled **“The Summary of Our Responses”** and our responses to the other reviewers.

---

### Official Review · Reviewer_rsjC · 2021-07-18

**Rating:** 6
**Confidence:** 4

**Summary:**

The method proposes a module to dynamically scale the tensors within a temporal action recognition model that can act as a drop-in addition to existing models. To achieve this, the per-frame input tensors are pooled into vectors, and a LSTM is used to condition the linear scaling of the tensor depending on the (avgpooled) representation of the current frame, but also previous predictions. This is also extended to include the predictions from the previous layer. There are also a couple of tweaks to the LSTM to improve performance.

**Limitations And Societal Impact:**

The authors do not discuss the limitations despite claiming to do so in the introduction and supplementary. In terms of ethical impact, they reply N/A and might be correct.

**Main Review:**

The paper becomes somewhat clear as you read through, but it is not easy to get into. The terminology and the way the explanations go don't let the reader get an early understanding of what it's been tackled. For example, reading through the abstract I honestly had no idea what the paper was actually proposing. Then there are a number of explanations and terminology that are not easy to get through their meaning (I still don't understand what's "meta" about the method).

The paper is somewhat novel. The mechanism used is indeed novel, but similar self-referential scaling modules are common for images and there are definitely multiple variants for video (something for example noted in the literature review). While the authors propose an LSTM-based methodology, some other works have proposed convolution-based approaches, but the global aim looks similar to me. Speaking in an abstract (a bit hand-wavy), you have a function that takes a sequence of input tensors and predicts linear scaling. That function, in this case, is an LSTM variant. Despite this, the variant is a valid one methodologically speaking, and in my opinion the approach is correct.

From the experimental point of view, experiments need to be more conclusive (and I know there are a lot of tables!). The main problems are a) On Kinetics400, the performance improvement seems to go down as the method is better (TSN+Ours is 2.7% better than TSN, but TSM+Ours is +1.1% better than TSM, and TDN+Ours is 0.71% better than TDN). b) I miss a more direct comparison with other methods with similar aim - I know the module improves over base architectures, but there are other alternatives to combine the base architectures with (e.g. section "action recognition with attentive models" lists a few). In order to be a useful module, you not only need to improve the base architecture, you also need to present an advantage over similar modules. c) FLOPs are ok (and the numbers look good), but timings would make a lot of sense as these modules are oftentimes quite sequential in nature and FLOPs do reflect particularly poorly what the impact is in practice (admittedly, this is not something common in the literature so it is hard to hold it against the authors).

Small details (no need to address them, just suggestions):

The ssv1/v2 table on the supplementary shoudl be upgraded into the main body, and maybe Table 4 is spurious and can be replaced instead?

The authors might want to revise a bit Efficient Spatial-Temporal Modelling. Some assertions do not sound great to me, e.g. l77 (mixing 2D and 3D convs) and l78 (two-paths networks are not designed for efficiency - slowfast adds the second path on top of a standard, relatively heavy backbone, so it is "efficient" in the sense that it doesn't add "too much" for the accuracy gain).

Why is the "meta" term used? wouldn't it make the paper easier to understand to drop it?

The paragraph starting on l209 is not particularly clear when explaining the positioning of the modules. Maybe the authors could revise this paragraph to make it easier to understand (Later the ablation study has a very good explanation, which is the one I used to understand it)



**Time Spent Reviewing:**

4

---

> ### Author Response · Authors · 2021-08-10
> **Responses to Official Review by Reviewer rsjC**
>
> Thank you so much for the constructive comments, and the recognition of the novelty and the correctness of our method. Please see our below responses to your concerns one by one.
>
> 1.**To your comments regarding the paper presentation** “The paper becomes somewhat  clear as you read through, but it is not easy to get into… don't let the reader get an early understanding of what it's been tackled…reading through the abstract I honestly had no idea what the paper was actually proposing…(I still don't understand what's "meta" about the method).”
>
> **Our responses**: **(1)** We are really sorry for unclear descriptions, explanations and terminology in our paper, which will be revised to be more clear during paper revision. **(2)** According to your comments of **“Summary:”**, your understanding of our method is basically accurate. The key motivation of our method is to augment the spatial-temporal modeling ability of any deep action recognition model via improving linear transform parameters of its normalization layers. Prevailing normalization operators such as BN, GN and LN are not frame-adaptive (i.e., normalization parameters are fixed to feature maps of different frames at each layer during the inference) and ignore dependencies of the estimated normalization parameters between neighboring frames (at each layer) and between neighboring layers (with all frames). Therefore, the proposed method contains cross-temporal DNR (i.e., CT) and cross-layer DNR (i.e., CL) to handle the above problems. **(3)** Traditional “meta learning” requires extra training or fine-tuning procedure during the inference. For our DNR, we used the word “meta”|”meta learning” mainly referring to the dynamic strategy which utilizes a shared lightweight LSTM unit to generate frame-adaptive normalization parameters conditioned on the input priors (i.e., the current feature maps and the previous normalization parameters as discussed above). To a certain degree this is akin to Meta Networks [1] and HyperNetworks [2] which use a network to generate task-adaptive weights for another network. We sincerely thank you for pointing out this potentially misleading issue of the word “meta”|”meta learning”, and we will drop it and replace it by “frame-adaptive|attentive learning” during paper revision.
>
> 2.**To your comments regarding the novelty of our method and its differences against other existing self-referential attentive modules for video action recognition** “The paper is somewhat novel. The mechanism used is indeed novel…While the authors… some other works have proposed convolution-based approaches, but the global aim looks similar to me. Speaking…, is an LSTM variant. Despite this, the variant is a valid one methodologically speaking, and in my opinion the approach is correct.”
>
> **Our responses**: **(1)** Thanks for recognizing the novelty and the correctness of our method. **(2)** We agree with your comment “The mechanism used is indeed novel, but similar self-referential scaling modules are common for images and there are definitely multiple variants for video”. Actually, we discussed them in the Sections of Related Work (see Line#97 to Line#108 and Line#124 to Line#130) and Introduction (see Line#37 to Line#48). **(3)** Compared to existing self-referential scaling modules such as TEINet [32], TEA [29], TANet [33] and TDN [55], our DNR is different both in design focus and formulation, though all methods globally aim to improve the spatial-temporal modeling of deep action recognition models. Specifically, our method focuses on improving the normalization layers, while they focus on improving the building blocks of a deep action recognition model with different attentive module designs. More importantly, as discussed in our above responses to address the presentation issue, our method introduces two dynamic normalization and relay designs (CT and CL) learnt with a shared LSTM unit, which are the core contributions of our paper (stated in Line#160 to Line#162). **(4)** Due to different design focus and formulation, our method can be applied to these existing temporal modules and show improved model accuracy, as can be seen from results shown in Table 6 of the main paper and Table 2 of the supplementary materials.
>
> 3.**To your comments regarding three main problems to the experiments** “From the experimental point of view, experiments need to be more conclusive…The main problems are a)…. b)…. c)…(admittedly, this is not something common in the literature so it is hard to hold it against the authors).”
>
> **Our responses**: **(1)** To the first problem “a)…”, yes, the performance improvement over different baselines is not at the same level. On Kinetics400, the performance improvement of our DNR indeed tends to go down as the baseline model becomes better (actually this is something common in addressing visual recognition tasks with deep CNNs, e.g., see Table 1 of SENet [19] for similar experimental results): TSN+DNR is 2.7% better than TSN (71.65% top-1 accuracy), TSM+DNR is 1.1% better than TSM (73.78% top-1 accuracy), TANet+DNR is 0.65% better than TANet (76.28% top-1 accuracy), and TDN+DNR is 0.71% better than TDN (76.34% top-1 accuracy). Note these four baseline models contain three types of advanced action recognition models with 2D CNNs. Specifically, TSN [56] is a popular temporal module, TSM [31] uses an improved temporal module with shift operations across neighboring frames, while more advanced attentive temporal modules are used in TANet [33] (built with dynamic convolutions) and TDN [55] (encoding frame differences into the attentive design). Particularly, TANet and TDN are two of the latest top-performing 2D attentive temporal modules, in this point of view, the performance improvement of our DNR is decent as DNR is merely added to normalization layers. **(2)** To the second problem, besides four basic backbones (namely BNIception, ResNet50, ResNet101 and ResNeXt101, see Table 2), as clarified in the above responses, we actually considered three types of temporal module (where TANet [33] and TDN [55] are two of the latest top-performing attentive modules) as our baselines. In Table 1d, a comparison of our relay design with SENet is also provided. Another experimental comparison of our method and state-of-the-art methods on Sth-Sth V1&V2 is provided in Table 2 of the supplementary materials. **(3)** To the third problem of using FLOPs to compare the computational cost of different models, we agree that the runtime inference speed is more critical for real deployment of action recognition models. You are referred to **our second set of responses to Reviewer zNtD** for the reason why we originally reported FLOPs instead of runtime inference speed to compare computational cost of different methods on Kinetics-400, and an extra comparison of runtime inference speed of different baseline models w/o and w/ our DNR.
>
> 4.**To the limitations of our method**, you are referred to **our seventh set of responses to Reviewer kHhy** for detailed discussions.
>
> 5.**To your suggestions** in **“Small details (no need to address them, just suggestions)”**, we really appreciate these constructive suggestions and would be happy to revise our paper following them accordingly. **Specifically**, **(1)** We will replace the Sth-Sth V1&V2 part of Table 4 in the main paper by Table 2 in the supplementary materials. **(2)** Yes, mixing 2D and 3D convolutions as well as two-path networks are not really efficient designs especially compared to 2D counterparts, they are “efficient” to their own motivations. We will revise related descriptions accordingly. **(3)**  We will drop the word “meta”, making the paper easier to understand. **(4)** We will revise the paragraph starting on Line#209 using the explanation in the ablative study, making it easier to understand.
>
> **Finally**, regarding more improvements to the explanations, the discussions and the experiments that we have made, you are referred to our top-level comments titled **“The Summary of Our Responses”** and our responses to the other reviewers. We will carefully revise the manuscript of our work w.r.t. insightful comments by all reviewers and our responses.
>
> [1] D. Ha, A. M. Dai, and Q. V. Le. Hyperneteworks. In ICLR, 2017.
>
> [2] T. Munkhdalai and H. Yu. Meta networks. In ICML, 2017.

---

> > ### Comment · Reviewer_rsjC · 2021-08-19
> > **Thanks for such a detailed response**
> >
> >
> > I have to say that the rebuttal is very strong and I see some of the mistakes I had in my initial understanding. I think the main aspect is, the authors show that the module stacks up with other attentive mechanism rather than trying to compare them more directly, which is fine.
> >
> > More in detail:
> >
> > 1) Paper presentation issues are hard for rebuttals, besides stating the intention to improve it, so I'll move on.
> > 2) Ok, I can see why the aim is different...
> > 3) I see that the main point for TANet and TDN is that the performance improvements stack up with those of the corresponding self-referential attentive modules. Happy with that. The performance improvement over TSM is though a bit limited while there is no "stack-up" effect (no other similar module used here). I can see though that the timings are actually quite encouraging, I thought it was going to be a worse impact in terms of latency.
> > 4 + 5 are ok too. Reply to 5 was really not needed (thanks anyway for it).
> >
> > All in all, I think the rebuttal has been strong and I'm happy to raise my score. I see how some the issues I raised were me not understanding what was meant in the experimental result design clearly, and also an important issue has been addressed with the timings.
> >
> > The only thing I'm still curious about is why the extra latency incurred when using DNR is small for TSN/TSM and much larger for TANet and TDN?

---

> > > ### Author Response · Authors · 2021-08-20
> > > **Responses to the Extra Question by Reviewer rsjC**
> > >
> > > Thank you so much for new informative comments. We are so glad that our responses to your concerns/questions are all well accepted by you.
> > >
> > > **To your extra question** “The only thing I'm still curious about is why the extra latency incurred when using DNR is small for TSN/TSM and much larger for TANet and TDN?”
> > >
> > > **Our responses** are: **(1)** Indeed, the **absolute** extra latency cost (AELC) incurred when using our DNR is small for the baseline TSN/TSM and is much larger for the baseline TANet/TDN, but the **relative** extra latency cost (RELC) is at a similar level and shows a decreased trend. Specifically, according to **the table shown in our second set of responses to Reviewer zNtD**, under the standard test settings of 8$\times$10$\times$3 sampled frames per video and a ResNet50 backbone, the AELC from our DNR is 0.018/0.014 (second per video) for TSN/TSM vs. 0.043/0.050 (second per video) for TANet/TDN (about 3$\times$ larger than that for TSN/TSM), and the RELC from our DNR is 19.6%/13.3% (second per video) for TSN/TSM vs. 12.0%/10.5% (second per video) for TANet/TDN (much lower than that for TSN/TSM). **(2)** The reasons for above observations are in **two aspects of architectural temporal modeling designs in these four baselines**. **Firstly**, most of recent advanced temporal modules for action recognition, including TSM [31], TANet [33] and TDN [55], adopt the seminal temporal segment design (with RGB input frames) proposed in TSN [56]. That is, TSN is adopted as the very basic structure, and hence it is the fastest one among our four baselines. **Secondly, compared to TSN**, TSM additionally adds temporal shift operations (which are well optimized in the public TSM code used in our benchmarking) to input feature maps of the first convolutional layer of every residual blocks of the backbone network, leading to a slightly lower runtime speed (0.105 vs. 0.092 second per video); TANet introduces a temporal adaptive module (a two-branch design with dynamic convolutions to extract local and global temporal features) into each residual block of the backbone network, leading to a significantly lower runtime speed (0.357 vs. 0.092 second per video); based on a temporal difference operator applied over sparsely sampled frames from multiple segments, TDN introduces a short-term temporal difference module and a long-term temporal difference module to residual blocks of the first two stages and the other three stages of the backbone network respectively, leading to the lowest runtime inference speed (0.476 vs. 0.092 second per video). Summarily, TSM does not change the number of convolutional layers in the very basic TSN design with a given backbone network, but TANet/TDN introduces a significantly increased number of convolutional layers to TSN. **Recall that** our DNR is merely applied to normalization layers conventionally added after convolutional layers, in this circumstance, its corresponding extra latency cost is directly related to the number of convolutional layers of the baseline model. Therefore, the result trends (including your observed one) described in our first response are observed.
> > >
> > > We hope our above responses are helpful to address your extra question. Thanks again for your very thorough and constructive comments.

---

### Author Response · Authors · 2021-08-10
**The Summary of Our Responses to All Official Reviews**

Dear Reviewers, Area Chairs and Program Chairs,

We sincerely thank all three reviewers for their thorough and constructive comments, and the recognition of the novelty of our work.

In the past week, we carefully improved the explanations, the discussions, and the experiments of our work to address the concerns and requests by all reviewers. **Summarily, we made following improvements**: **(1)** To the potentially misleading term “meta”/“meta learning” (concerned by Reviewer rsjC and Reveiwer kHhy), we will drop it and replace it by “frame-adaptive|attentive”/” frame-adaptive|attentive learning” according to the basic insight of our dynamic video normalization and relay design. **(2)** To the experiments section, under the exactly same settings we provide a runtime inference speed comparison of our method and its counterparts (a common concern from all three reviewers) first, and then more results as well as explanations are also provided. **(3)** To the motivation of our method and the differences of our method against other existing self-referential attentive modules for video action recognition (concerned by Reviewer rsjC), we provide more clear clarifications. **(4)** To the limitations of our method, we follow the insightful suggestions by Reveiwer kHhy, and provide related discussions. **(5)** We also provide detailed responses to the other questions/suggestions raised by each reviewer one by one.

We hope our detailed responses are helpful to address the concerns, questions and requests of all reviewers. **Finally**, we will carefully revise the manuscript of our work w.r.t. insightful comments by all reviewers and our responses.

---

### Decision · Program_Chairs · 2021-09-27

**Decision:**

Accept (Poster)

**Comment:**

This paper presents work on action recognition.  The main contribution is a drop-in module that dynamically scales tensors.  The initial reviews pointed to concerns over clarity, novelty, and experimetns.  The reviewers engaged in discussion and were satisfied with the authors' responses to their initial concerns.  The reviewers unanimously recommended acceptance of the paper after the discussion period, based on the novel approach to dynamic scaling for videos.  The authors are encouraged to make changes to presentation to help in describing the contributions in a clearer fashion, especially earlier in the paper, to improve its readability and future impact.